# Prenatal Iron Deficiency and Choline Supplementation Interact to Epigenetically Regulate *Jarid1b* and *Bdnf* in the Rat Hippocampus into Adulthood

**DOI:** 10.3390/nu13124527

**Published:** 2021-12-17

**Authors:** Shirelle X. Liu, Amanda K. Barks, Scott Lunos, Jonathan C. Gewirtz, Michael K. Georgieff, Phu V. Tran

**Affiliations:** 1Department of Pediatrics, University of Minnesota, Minneapolis, MN 55455, USA; liu00459@umn.edu (S.X.L.); barks012@umn.edu (A.K.B.); georg001@umn.edu (M.K.G.); 2Department of Psychology, University of Minnesota, Minneapolis, MN 55455, USA; jgewirtz@umn.edu; 3Biostatistical Design and Analysis Center, Clinical and Translational Science Institute, University of Minnesota, Minneapolis, MN 55455, USA; slunos@umn.edu

**Keywords:** *Bdnf*, choline, epigenetics, gene expression, histone modification, iron deficiency, *Jarid1b*

## Abstract

Early-life iron deficiency (ID) causes long-term neurocognitive impairments and gene dysregulation that can be partially mitigated by prenatal choline supplementation. The long-term gene dysregulation is hypothesized to underlie cognitive dysfunction. However, mechanisms by which iron and choline mediate long-term gene dysregulation remain unknown. In the present study, using a well-established rat model of fetal-neonatal ID, we demonstrated that ID downregulated hippocampal expression of the gene encoding JmjC-ARID domain-containing protein 1B (JARID1B), an iron-dependent histone H3K4 demethylase, associated with a higher histone deacetylase 1 (HDAC1) enrichment and a lower enrichment of acetylated histone H3K9 (H3K9ac) and phosphorylated cAMP response element-binding protein (pCREB). Likewise, ID reduced transcriptional capacity of the gene encoding brain-derived neurotrophic factor (BDNF), a target of JARID1B, associated with repressive histone modifications such as lower H3K9ac and pCREB enrichments at the *Bdnf* promoters in the adult rat hippocampus. Prenatal choline supplementation did not prevent the ID-induced chromatin modifications at these loci but induced long-lasting repressive chromatin modifications in the iron-sufficient adult rats. Collectively, these findings demonstrated that the iron-dependent epigenetic mechanism mediated by JARID1B accounted for long-term *Bdnf* dysregulation by early-life ID. Choline supplementation utilized a separate mechanism to rescue the effect of ID on neural gene regulation. The negative epigenetic effects of choline supplementation in the iron-sufficient rat hippocampus necessitate additional investigations prior to its use as an adjunctive therapeutic agent.

## 1. Introduction

Iron deficiency (ID) is a common micronutrient deficiency worldwide, disproportionately affecting 40–50% of pregnant women and preschool-aged children in middle-to-low-income countries [1,2]. Human studies show an association between early-life ID and persistent neurodevelopmental effects in formerly iron-deficient individuals in adolescence [3] and adulthood [4]. Low maternal iron intake during pregnancy is also associated with increased risk for schizophrenia [5] and autism [6] in the offspring. Collectively, early-life ID can produce lasting negative effects on neurocognitive function and adult mental health. These long-term neurological effects are also observed in animal models of fetal-neonatal ID. In animal models, ID during the fetal and early postnatal periods compromises neurodevelopment [7,8,9]. Brain development during fetal and early postnatal life is an energy-intensive process [10,11,12] and is disrupted in a rat model of fetal-neonatal ID [13,14,15]. In addition, formerly iron-deficient rats show long-term behavioral deficits in hippocampal-dependent learning and memory [16,17,18] associated with altered expression of genes critical for neuron development [14,19,20] and gene networks implicated in schizophrenia, autism and mood disorders [21].

The presence of long-term behavioral abnormalities in formerly iron-deficient humans and rodent models despite early treatment with iron following diagnosis indicates the need to understand the mechanisms underlying the effects to ultimately identify adjunct treatments for early-life ID. One attractive candidate is maternal choline supplementation. Choline is an essential nutrient that is critically involved in early brain development [22,23], and is readily found in over 630 food sources [24], which makes it more accessible than iron [25]. Choline supplementation during specific pre- and postnatal time windows reverses some cognitive deficits observed in a rodent model of fetal-neonatal ID associated with increased expression of neurotrophic factors such as brain-derived neurotrophic factor (BDNF) [21,26,27]. However, the mechanism by which choline reverses these effects is unknown. One possibility is that choline acts as a methyl donor and thereby modifies the epigenetic landscape through DNA methylation and histone methylation [28,29].

Epigenetic modifications can contribute to the long-term effects of fetal-neonatal ID on neural gene dysregulation [30,31]. Of particular interest is the family of iron-containing JmjC-ARID (JARID) domain containing dioxygenases, which require iron as a cofactor to remove methyl groups from lysine residues of histones [32]. Previously, we demonstrated that persistent dysregulation of JARIDs was accompanied by changes in histone methylation at the *Bdnf* locus in adult hippocampus following early-life ID [31,33]. In the present study, we assessed the regulatory effects of fetal-neonatal ID and prenatal choline supplementation through a hypothesized iron-mediated epigenetic mechanism involving JmjC-ARID domain-containing protein 1B (JARID1B) and its target *Bdnf* in the rat hippocampus across postnatal ages. Although ID also reduces expression of other iron-containing JARIDs [33], we elected to analyze JARID1B due to its function in demethylating tri- and di-methylated histone H3K4, which is canonically associated with transcriptional activation, especially of *Bdnf*, and its important regulatory effects in neural development [32,34]. We hypothesized that ID dysregulates *Jarid1b* expression, consequently modifying epigenetic signatures at the *Bdnf* promoters and that prenatal choline supplementation mitigates these regulatory changes.

## 2. Materials and Methods

### 2.1. Animals

Gestational day (G) 2 pregnant Sprague–Dawley rats were purchased from Charles River Laboratories (Wilmington, MA, USA). Rats were maintained in a 12-h:12-h light/dark cycle with ad lib food and water. ID was induced by dietary manipulation as described previously [26,27,35]. In brief, to generate iron-deficient pups, pregnant dams were given a purified iron-deficient diet (4 mg Fe/kg, TD.80396; Harlan Teklad) from G2 to postnatal day (P) 7 when lactating dams were switched to a purified iron-sufficient diet (200 mg Fe/kg, TD.09256; Harlan Teklad). Control iron-sufficient pups were generated from pregnant dams maintained on the iron-sufficient diet. Both diets were similar in all respects with the exception of the iron (ferric citrate) content. Details of diet contents have been described in our previous study [26]. Half of the dams on an iron-sufficient or iron-deficient diet received dietary choline supplementation (5.0 g/kg choline chloride supplemented [36,37]; iron-sufficient with choline: TD.1448261, iron-deficient with choline: TD.110139; Harlan Teklad) from G11-18, while the other half of the dams received their iron-sufficient or iron-deficient diet with standard choline content (1.1 g/kg). Thus, dams and their litters were assigned to one of four groups based on maternal diet: iron-deficient with G11-18 choline supplementation (IDCh), iron-deficient without supplemental choline (ID), always iron-sufficient with G11-18 choline supplementation (ISCh), always iron-sufficient without supplemental choline (IS) (Figure 1). All litters were culled to 8 pups with six males and two females at birth. Only male offspring were used in experiments, because all prior extant data on long-term effects are in males [21,27,30,31]. To avoid litter-specific effects, two male rats per litters and ≥3 L/group were used in the experiments. The University of Minnesota Institutional Animal Care and Use Committee approved all experiments in this study (Protocol # 2001-37802A).

### 2.2. Hippocampal Dissection

Rats were euthanized by injection of pentobarbital (100 mg/kg, intraperitoneal). Brains were removed and bisected along the midline on an ice-cold metal block. Hippocampus from P0 (end of proliferation), P15 (start of robust dendrite differentiation), and P65 (mature) rat was dissected and immediately flash-frozen in liquid nitrogen and stored at −80 °C for further use.

### 2.3. RNA Isolation and cDNA Synthesis

Total RNA was isolated from 1 hippocampal lobe per rat using RNAqueous Total RNA Isolation Kit (Invitrogen). cDNA synthesis was performed using 1 µg isolated RNA and the High-Capacity RNA-to-cDNA Kit (Applied Biosystems), as previously described [7].

### 2.4. Nuclear Protein Isolation

Nuclear proteins were isolated from hippocampal tissue using the Epiquik Nuclear Extraction Kit II (Epigentek), as per the manufacturer’s protocol. Protein concentration was determined by a standard Bradford protein assay using serial dilutions of bovine serum albumin (BioRad) as standards. 

### 2.5. JARID Activity

JARID1 enzymatic activity was assessed from 20 µg of nuclear protein using the Epigenase JARID Demethylase Activity Assay Kit (Epigentek), following the manufacturer’s protocol. This assay quantified total JARID1, including JARID1A, 1B, 1C, and 1D, demethylase activity. In this assay, histone H3K4me3 substrate was coated onto microplate wells. Active JARID1 bound and removed the methyl group from the substrate. Demethylated products were recognized by a specific antibody, which was quantified by colorimetric absorbance at 450 nm (Epigentek). JARID1 activity was quantified in the hippocampus from 6 animals per treatment group. All samples were analyzed in technical duplicate. Results were calculated from the average of the duplicates.

### 2.6. Western Blot

Western blot analysis was used to quantify levels of JARID1B using a previously described protocol [35]. In brief, 30 µg of hippocampal protein lysate were separated using a 4–20% SDS-PAGE gel (Novex, Life Technologies). Proteins were blotted onto a nitrocellulose membrane (Pierce Protein Biology), blocked with Blocking Buffer for Fluorescent Western Blotting (Rockland Immunochemicals), and incubated with antibodies against JARID1B (Abcam) and beta actin (Sigma). Following PBS + 0.1% Tween–20 washes, blots were incubated in Alexa–700 anti-mouse (Invitrogen) and IR Dye-800 anti-rabbit (Rockland) and scanned by Near-Infrared Fluorescent using the Odyssey Infrared Imaging System (Li-Cor Biosciences). Integrated intensity of protein bands normalized to beta actin was determined using Photoshop CS 4 (Adobe, San Jose, CA, USA).

### 2.7. Chromatin Immunoprecipitation (ChIP) Assay

ChIP experiments were performed as previously described [31]. In brief, chromatin was prepared from hippocampal tissue following the manufacturer’s recommendation (Millipore). Hippocampi were homogenized in ice-cold PBS (500 µL). Chromatin was cross-linked in 1% formaldehyde solution (Sigma), pelleted by centrifugation, resuspended in 500 µL lysis buffer (1% SDS, 10 mM EDTA, 50 mM Tris pH 8.1, 1 mM PMSF, 10 µL 10× protease inhibitor cocktails (Roche)), and fragmented by sonication (Bioruptor Pico, Diagenode). Chromatin fragments were visualized by agarose gel electrophoresis following a cross-linking reversal (0.2 M NaCl, 65 °C overnight) to ensure optimal fragment sizes. Fragmented chromatin was diluted 10-fold with ChIP dilution buffer (0.01% SDS, 1.1% Triton X-100, 1.2 mM EDTA, 16.7 mM Tris-pH 8.1, 167 mM NaCl), pre-cleared with 75 µL of Protein A agarose (50% slurry, Sigma), and immunoprecipitated by ChIP-grade antibody (4 °C, overnight). ChIP-grade antibodies used in this study included JARID1B (Abcam), histone deacetylase 1 (HDAC1) (Diagenode), H3K9ac (Diagenode), H3K9me3 (Diagenode), upstream stimulatory factor 1 (USF1) (Santa Cruz Biotechnologies), and phosphorylated cAMP response element-binding protein (pCREB) (Millipore Sigma). Normal IgG was used as a negative control. The antibody-histone complex was collected by the addition of 60 µL of Protein A agarose slurry with mixing (4 °C, ≥1 h) and subsequently eluted in 500 µL of elution buffer (1% SDS, 0.1 M NaHCO3) following buffer (as per the manufacturer’s protocol) rinses. Reverse cross-linking was achieved by incubation in NaCl (0.2 M, 65 °C, overnight). A protease digestion (20 µg proteinase K, 20 mM EDTA, 100 mM Tris-pH 6.5, 45 °C, 1 h) was performed to recover DNA, which was further purified using phenol/chloroform extraction and ethanol precipitation (0.1 × 3 M sodium acetate, pH 5.2, 2 × 100% ethanol). Levels of enriched *Gapdh* (active) and *Myod1* (inactive) loci were used to validate ChIP experiments.

### 2.8. Real-Time Quantitative PCR (RT-qPCR)

For mRNA quantitation, TaqMan Universal PCR Master Mix (Applied Biosystem) was used with gene expression assays (Applied Biosystem; *G9a*, Assay ID: Rn01758882_m1; *Jarid1b*, Assay ID: Rn01758882_m1; *Suv39h1*, Assay ID: Rn01528294_g1). Genes that encode TATA-box binding protein (*Tbp*, Assay ID: Rn01455646_m1) or ribosomal protein s18 (*Rps18*, Assay ID: Rn01428915_g1), the expressions of which were not altered by ID, were used as normalizers. For analysis of precipitated DNA from ChIP experiments, Fast SYBR green master mix (Applied Biosystem) was used to amplify the *Bdnf4*, *Bdnf6*, and *Jarid1b* promoters using validated primers (*Bdnf4*, For-TCGAGGCAGAGGAGGTATCA, Rev- CCTCTCCTCGGTGAATGGGA; *Bdnf6*, For-GATGAAAGGTTTGGCTTCTGTG, Rev- TCGGTGAATGGGAAAGTGG; *Jarid1b*, For-TGGGAGAGTTCCTGCCT, Rev- GCCGGATCTTGTGGATGAA). Input DNA (10%) was used as a normalizer to account for input amount (ΔCt). Data were expressed as fold change (2-ΔΔCt) relative to IS control using one of the IS samples as a calibrator (ΔΔCt). Real-time PCR was performed with QuantStudio™ 3 (Thermo Fisher, Waltham, MA, USA).

### 2.9. Experimental Design and Statistical Analysis

For data that were collected from IS, ID, ISCh and IDCh groups, a two-way ANOVA with interaction (iron status × choline supplementation) was used at each time point. If there was a significant interaction, the simple effect of iron status or choline supplementation was analyzed at each choline supplementation condition or iron status, respectively. Completed two-way ANOVA results are shown in Table 1. α was set at 0.05 for all comparisons. Data were graphed and statistically analyzed using GraphPad Prism 9 (GraphPad Software, Inc., San Diego, CA, USA) and SAS V9.4 (SAS Institute Inc., Cary, NC, USA).

## 3. Results

### 3.1. Iron Deficiency Downregulates Hippocampal Jarid1b Expression

To determine the effects of ID on *Jarid1b* expression across critical periods of hippocampal development, *Jarid1b* mRNA levels were compared among IS, ID, ISCh and IDCh groups. An interaction between iron status and choline supplementation on *Jarid1b* expression was found at P15 (*p* = 0.0341) and P65 (*p* = 0.0313). Compared to the iron-sufficient group, the effects of iron status and choline supplementation were significant at P15 (ID vs. IS, *p* = 0.0003; ISCh vs. IS, *p* = 0.0096). At P65, the effect of iron status and choline supplementation in the iron-deficient groups remained significant (IS vs. ID, *p* = 0.0029; IDCh vs. ID, *p* = 0.0051). Compared to the iron-sufficient control group, hippocampal *Jarid1b* expression was lower in the iron-deficient group at P15 (Figure 2B, ID vs. IS) and P65 (Figure 2C, ID vs. IS). Prenatal choline supplementation ameliorated the effect of ID at P65 (Figure 2C, IDCh vs. ID), but not at P15 (Figure 2B, IDCh vs. ID, *p* = 0.7). Choline supplementation also reduced *Jarid1b* expression in the P15 ISCh group (Figure 2B, ISCh vs. IS). Western blot analysis corroborated the lower hippocampal *Jarid1b* expression caused by ID at P15 and P65 (Figure 2D, ID vs. IS). While we could not specifically determine JARID1B activity, total JARID1 (JARID1A, B, C, and D) enzymatic activity was assessed and showed a lower activity level in the P15 ID group and a higher activity level in the P65 iron-replete ID group (Figure 2E, ID vs. IS).

### 3.2. Iron Deficiency and Prenatal Choline Supplementation Alter Jarid1b’s Epigenetic Signatures

We then tested whether ID and prenatal choline supplementation were associated with epigenetic regulation at the *Jarid1b* promoter. Since histone deacetylase 1 (HDAC1) has been reported to downregulate *Jarid1b* expression [38], HDAC1 enrichment was compared among the 4 groups. An interaction between iron status and choline supplementation on HDAC1 enrichment was found at P15 (*p* = 0.0012). The effects of iron status and choline supplementation were significant (Figure 3B; ID vs. IS, *p* = 0.0194; IDCh vs. ISCh, *p* = 0.0086; ISCh vs. IS, *p* = 0.0051; IDCh vs. ID, *p* = 0.0363). No effects were found at P0 or P65 (Figure 3A,C). The level of HDAC1 enrichment was greater in the P15 iron-deficient group (Figure 3B, IS vs. ID). Choline supplementation ameliorated the effect of ID (Figure 3B, IDCh vs. ID) and induced a higher level of HDAC1 enrichment in the ISCh group (Figure 3B, ISCh vs. IS). These findings were further corroborated by the concomitant changes in H3K9 acetylation (H3K9ac). An interaction between iron status and choline supplementation on H3K9ac enrichment was found at P15 (*p* = 0.0331), where both ID and choline supplementation reduced H3K9ac enrichment at P15 (Figure 3E; ID vs. IS, ISCh vs. IS). At P65, only a main effect of choline supplementation was found with a reduced H3K9ac enrichment (Figure 3F).

We further tested whether the changes in HDAC1 recruitment and histone H3 acetylation would reduce the recruitment of phosphorylated cAMP response element-binding protein (pCREB), which can reflect neuronal activity [39,40,41]. While no clear effect was found at P0 or P15 (Figure 3G,H), a main effect of choline supplementation was found with a lower pCREB enrichment at P65 (Figure 3I).

### 3.3. Iron Deficiency and Prenatal Choline Supplementation Alter Hippocampal Histone H3K9 Methylation at the Jarid1b Locus

Given the documented effects of choline supplementation on expression of histone H3K9 methyltransferases and histone H3K9 methylation [29,42], we determined the effects of ID and prenatal choline supplementation on the hippocampal expression of H3K9 methyltransferases *G9a* and *Suv39h1*. For *G9a* expression, an interaction between iron status and choline supplementation was found at P0 (*p* = 0.0491). ID and choline supplementation induced a higher *G9a* expression in P0 hippocampus (Figure 4A; ID vs. IS, ISCh vs. IS). A main effect of iron status was found with a reduced *G9a* expression at P15 (Figure 4B). A main effect of choline supplementation was found with a lower *G9a* expression at P65 (Figure 4C). For *Suv39h1* expression, an interaction between iron status and choline supplementation was found at P0 (*p* = 0.0165). ID induced a higher *Suv39h1* expression and choline supplementation removed this effect (Figure 4D; ID vs. IS, IDCh vs. ID). An interaction between iron status and choline supplementation was also found at P65 (*p* = 0.0152), where choline supplementation reduced *Suv39h1* expression (Figure 4F; IDCh vs. ISCh, IDCh vs. ID). The main effects of both iron status and choline supplementation were found at P15, where ID reduced and choline supplementation induced *Suv39h1* expression (Figure 4E).

Since both G9A and SUV39H1 target histone H3K9 [43], we quantified H3K9me3 enrichment at the *Jarid1b* promoter by quantitative chromatin immunoprecipitation (qChIP) to determine the relevance of the changes in histone methyltransferase expression in the ID hippocampus. No clear difference was found among comparison groups at P0 or P15 (Figure 4G,H). An interaction between iron status and choline supplementation (*p* = 0.0081) was found at P65. ID induced a higher H3K9me3 enrichment (Figure 4I, ID vs. IS) and a choline supplementation ameliorated the effect of ID (Figure 4I, IDCh vs. ID).

### 3.4. Iron Deficiency and Choline Supplementation Alter Histone Modification at the Bdnf Promoters

To further analyze how fetal-neonatal ID alters the epigenetic regulation of *Bdnf*, a target of JARID1B [31,44,45,46,47,48,49,50], we quantified changes in histone modifications and binding of known transcription factors at the *Bdnf4* and *Bdnf6* promoters by qChIP. Initial studies between ID and IS groups not supplemented with choline showed that, during the peak of hippocampal ID at P15 [51], enrichment levels of H3K4me3 were lower at both the *Bdnf4* and *Bdnf6* promoters (Figure 5A and Figure 6A), which was accompanied by a lower enrichment of upstream stimulating factor 1 (USF1) and a higher JARID1B enrichment at the *Bdnf6* promoter (Figure 6A). These epigenetic marks were not different between P65 iron-replete ID and IS groups (Figure 5B and Figure 6B). Subsequent analyses revealed a main effect of iron status with a lower H3K9ac enrichment at the *Bdnf4* promoter at P15 (Figure 5C). A main effect of choline supplementation on H3K9ac enrichment at the *Bdnf4* promoter was also found at P65 (Figure 5D). Analysis of pCREB enrichment at the *Bdnf4* promoter found an interaction between iron status and choline supplementation (*p* = 0.0417). ID and choline supplementation lowered pCREB enrichment at P15 (Figure 5E; ID vs. IS, ISCh vs. IS). A main effect of choline supplementation was found with a lower pCREB enrichment at P65 (Figure 5F).

The increased JARID1B enrichment prompted us to assess HDAC1 recruitment at the *Bdnf6* promoter. An interaction between iron status and choline supplementation was found with HDAC1 enrichment at P15 (*p* = 0.0199) and P65 (*p* = 0.0063). At P15, ID induced a higher HDAC1 enrichment and choline supplementation removed this effect (Figure 6C; ID vs. IS, IDCh vs. ID). Similar effects were found for HDAC1 enrichment at P65 (Figure 6D; ID vs. IS, IDCh vs ID). Despite these changes, only the main effect of iron status was found with a lower H3K9ac enrichment at P15 (Figure 6E). At P65, a main effect of choline supplementation was found with lower H3K9ac (Figure 6F) and pCREB (Figure 6J) enrichments. A main effect of iron status was also found with a reduced pCREB enrichment at P65 (Figure 6J).

## 4. Discussion

Long-term gene dysregulation, particularly of the *Bdnf* [49] in adult rat hippocampus [20,31], suggests a possible epigenetic mechanism whereby early-life iron ID exerts an enduring influence into adulthood. Nevertheless, the link between ID and *Bdnf* dysregulation remains unknown. The present study provides evidence that early-life ID alters the regulation and enzymatic activity of JARID1B, an iron-dependent histone demethylase. These changes could contribute to ID-altered epigenetic regulation of the *Bdnf* gene, which is a known target of JARID1B [31,44,45,46,47,48,50]. The current findings provide an iron-specific mechanism to underlie the neural gene dysregulation caused by ID in the developing hippocampus.

JARID1B is an iron-containing demethylase that removes methyl groups from histone H3K4me2/3 and plays a role in cellular oxygen sensing [52,53,54]. Hippocampal JARID1B expression was lowered during (P15) and beyond (P65) the period of ID, the latter indicating a long-term change in its regulation mediated by an epigenetic modification of the *Jarid1b* gene. The early-life downregulatory effect might function as an adaptive response to low oxygen or reduced energetic environment caused by ID to inappropriately alter expression of genes regulating neural growth and differentiation [55,56,57]. Consequently, this adaptation could lead to an accelerated hippocampal aging [19], while attempting to improve the fitness of the animal beyond the fetal period [11]. Mechanistically, a greater HDAC1 recruitment and lower H3K9 acetylation could only account for *Jarid1b* downregulation in the iron-deficient hippocampus (Figure 3D,E). The mechanism of long-term *Jarid1b* downregulation likely involves additional epigenetic modifications (e.g., H3K27me3) and requires further investigation. The recovery of overall JARID1 activity with iron treatment across postnatal ages, despite the persistent *Jarid1b* downregulation suggests that iron therapy following a period of ID could potentially restore iron-containing JARID by incorporating the appropriate metal [58,59]. However, we speculate that the window of opportunity for such restoration may be time-limited during hippocampal development [13].

Given the documented beneficial effects of prenatal choline supplementation on normalizing gene expression (e.g., *Bdnf*) and neurocognitive function in adult rats that were iron-deficient during the fetal-neonatal period [21,26], it is not surprising that choline supplementation normalized *Jarid1b* long-term downregulation in the formerly iron-deficient adult rat hippocampus (Figure 2C, ID vs. IDCh). It is noteworthy that iron treatment alone was not sufficient to restore *Jarid1b* expression. These findings provide a functional link between this methyl diet and *Jarid1b* regulation under ID environment. The lower hippocampal expression of histone H3 methyltransferases, *G9a* and *Suv39h1*, and H3K9me3 enrichment at the *Jarid1b* promoter supports the epigenetic regulation of choline supplementation with long-term implications. Thus, the interaction between iron status and choline supplementation will need further investigation to optimize the utility of choline as an adjunctive therapy for early-life exposures.

Our previous findings showed that ID and choline supplementation epigenetically modified the *Bdnf* gene [31]. In the present study, we provide a potential mechanistic link between the iron-containing JARID1B histone demethylase, HDAC1 recruitment, and *Bdnf* gene regulation [44,45,46,47,48,60,61,62,63,64]. Our assessment of the effects of ID on the *Bdnf4* and *Bdnf6* (previously designated as *BdnfIII* and *BdnfIV* in rats, respectively) promoters revealed differential regulatory changes, with a greater short-term JARID1B binding and a corresponding lower H3K4me3 enrichment only at the *Bdnf6* promoter in the P15 ID hippocampus. In light of the concurrent lower *Jarid1b* expression and enzymatic activity, the findings suggest promoter-specific effects. Enrichment of a transcriptional repressive H3K9me3 marker was also different, with a lower level at the *Bdnf4* promoter and no change at the *Bdnf6* promoter. As such, both promoters were less transcriptionally poised or active in the iron-deficient hippocampus as evidenced by reduced enrichment of H3K9ac, pCREB, and USF1. Taken together, these findings suggest that the *Bdnf6* promoter, but not the *Bdnf4* promoter, was likely regulated by the iron-specific JARID1B mechanism. These epigenetic modifications were normalized in the P65 iron-replete ID hippocampus, indicating the short-term nature of both iron-dependent and -independent histone modifications in the developing ID rat hippocampus. These findings further add to the body of work demonstrating the dynamic and plastic properties of the hippocampus and epigenetic changes in response to external environments [65,66,67,68,69].

Choline supplementation is also known to modify histone methylation [29,42] and rescue long-term *Bdnf6* downregulation during fetal-neonatal ID [31]. However, here we provide evidence that choline supplementation to IS animals had a negative effect on histone H3K9 acetylation and pCREB enrichment at the *Bdnf* promoters. These findings suggest an interaction between choline and iron status in gene regulation during hippocampal development that is consistent with transcriptomic changes previously observed in the adult ISCh hippocampus [21]. In this context, the mode of choline action regulating specific histone modifications could involve distinct histone demethylases and methyltransferases identified in previous studies [29,42]. Alternatively, it is possible that the timing of analysis following choline supplementation to IS animals could account for the disparate findings in the present study and those previously reported [29,42], where analyses of histone methylation were performed in embryonic tissues immediately following the final day of choline supplementation. Likewise, the effects of choline supplementation on epigenetic regulation of *Bdnf* in the developing iron-deficient hippocampus could involve DNA methylation [70,71], which was not measured in the present study. 

In summary, the present study provides evidence for an epigenetic regulation of the iron-dependent mechanism that could underlie the previously documented dysregulation of the *Bdnf* gene in the adult rat hippocampus caused by early-life ID. Given that JARID1B plays a critical role in neural growth and differentiation [34,52,56], perturbed expression of this protein during neural development provides a molecular mechanism whereby early-life brain iron depletion alters the regulation of synaptic plasticity genes that lasts beyond the period of ID [14,19,20,21,72]. Changes in histone modifications associated with early-life ID or choline supplementation were dynamic and changeable across postnatal age in the rat hippocampus. Mechanisms underlying these dynamic changes in the nervous system may involve epigenetic modifiers similar to studies that assessed the relationship between the epigenome and postnatal life neuropathology such as Alzheimer’s disease, an associated risk of early-life ID anemia [73,74]. While there are substantial numbers of interactions among epigenetic factors in reprogramming gene expression, this study represents a novel assessment of the effects of iron, choline, and their potential interaction, laying down a foundation for additional study in this field.

## Figures and Tables

**Figure 1 nutrients-13-04527-f001:**
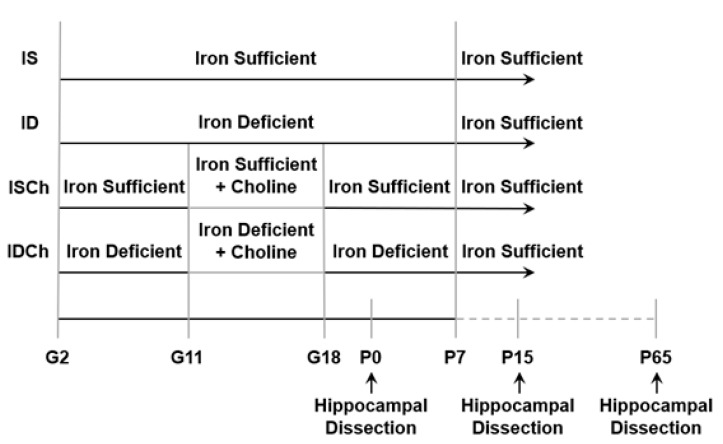
Graphical diagram of experimental groups. Abbreviations: Gestational day (G), iron-deficient group (ID), iron-deficient with choline supplementation group (IDCh), iron-sufficient group (IS), iron-sufficient with choline supplementation group (ISCh), and postnatal day (P).

**Figure 2 nutrients-13-04527-f002:**
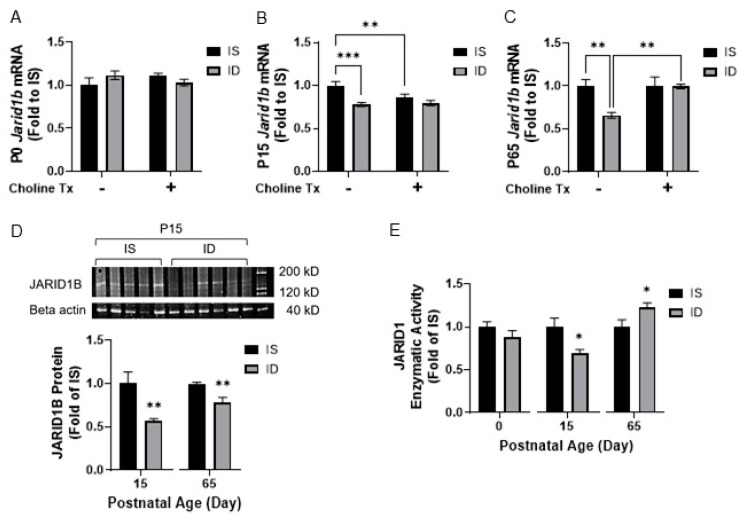
Iron deficiency alters expression and enzymatic activity of hippocampal JARID1 in the ID rats. (**A**–**C**) Hippocampal *Jarid1b* RNA expression in IS, ID, ISCh and IDCh groups at P0 (**A**), P15 (**B**) and P65 (**C**) quantified by Real-time quantitative PCR(RT-qPCR). (**D**) Hippocampal JARID1B protein expression in IS and ID groups at P15 and P65 measured with Western blot. Representative Western blot image of JARID1B at P15 is shown. (**E**) Hippocampal JARID1 activity in IS and ID groups at P0, P15 and P65 assessed with enzyme-linked immunosorbent assay. All data were normalized to the age-matched IS group. Tx: Treatment (Supplementation). Values are mean ± SEM; n = 5–6/group; * *p* < 0.05, ** *p* < 0.01, *** *p* < 0.001.

**Figure 3 nutrients-13-04527-f003:**
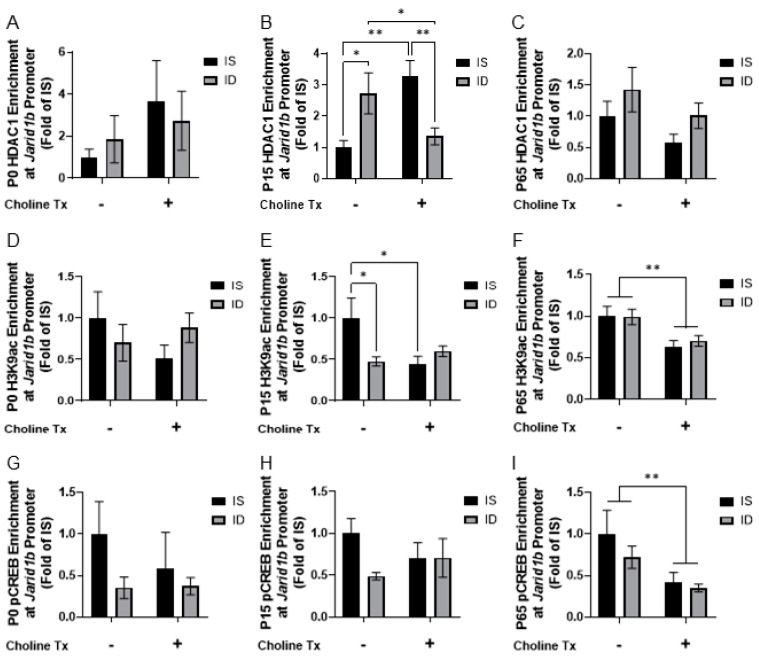
Iron deficiency and choline supplementation alter rat hippocampal *Jarid1b* regulation. (**A**–**C**) HDAC1 enrichment at the *Jarid1b* promoter in the 4 groups at P0 (**A**), P15 (**B**), and P65 (**C**). (**D**–**F**) H3K9ac enrichment at the *Jarid1b* promoter in the 4 groups at P0 (**D**), P15 (**E**), and P65 (**F**). (**G**–**I**) Phosphorylated cAMP response element-binding protein (pCREB) enrichment at the *Jarid1b* promoter in the 4 groups at P0 (**G**), P15 (**H**), and P65 (**I**). All data were assessed with quantitative chromatin immunoprecipitation (qChIP) and normalized to the age-matched IS group. Tx: Treatment (Supplementation). Values are mean ± SEM, n = 5–6/group, * *p* < 0.05, ** *p* < 0.01.

**Figure 4 nutrients-13-04527-f004:**
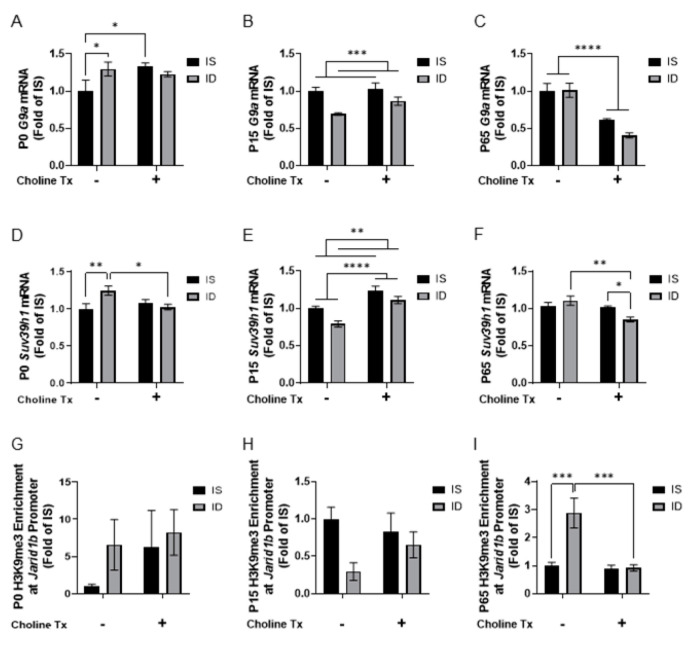
Iron deficiency and choline supplementation alter hippocampal histone H3K9 methylation. (**A**–**C**) Expression levels of *G9a* in the 4 groups at P0 (**A**), P15 (**B**) and P65 (**C**). (**D**–**F**) Expression levels of *Suv39h1* in the 4 groups at P0 (**D**), P15 (**E**) and P65 (**F**). (**G**–**I**) H3K9me3 enrichment at the *Jarid1b* promoter in the 4 groups at P0 (**G**), P15 (**H**) and P65 (**I**). All data were assessed with RT-qPCR or qChIP and normalized to the age-matched IS group. Tx: Treatment (Supplementation). Values are mean ± SEM; n = 5–6/group; * *p* < 0.05, ** *p* < 0.01, *** *p* < 0.001, **** *p* < 0.0001.

**Figure 5 nutrients-13-04527-f005:**
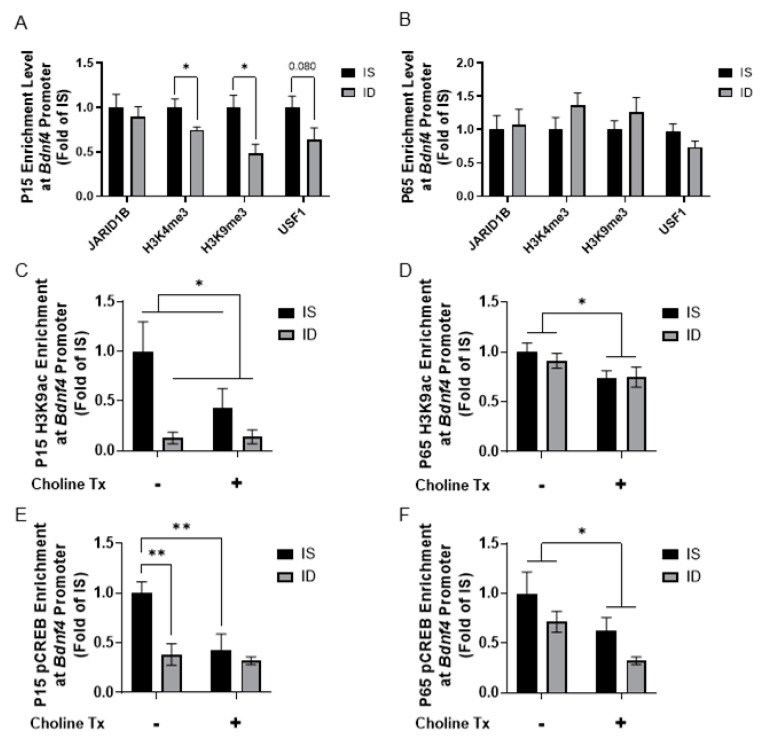
Iron deficiency and choline supplementation induce epigenetic changes at the hippocampal *Bdnf4* promoter. (**A**,**B**) Enrichment levels of JARID1B, H3K4me3, H3K9me3 and upstream stimulatory factor 1(USF1) at the *Bdnf4* promoter in P15 (**A**) and P65 (**B**) IS and ID groups. (**C**,**D**) H3K9ac enrichment at the *Bdnf4* promoter in the 4 groups at P15 (**C**) and P65 (**D**). (**E**,**F**) pCREB enrichment at the *Bdnf4* promoter in the 4 groups at P15 (**E**) and P65 (**F**). All data were assessed by qChIP and normalized to the age-matched IS group. Tx: Treatment (Supplementation). Values are mean ± SEM; n = 5–6/group; * *p* < 0.05, ** *p* < 0.01.

**Figure 6 nutrients-13-04527-f006:**
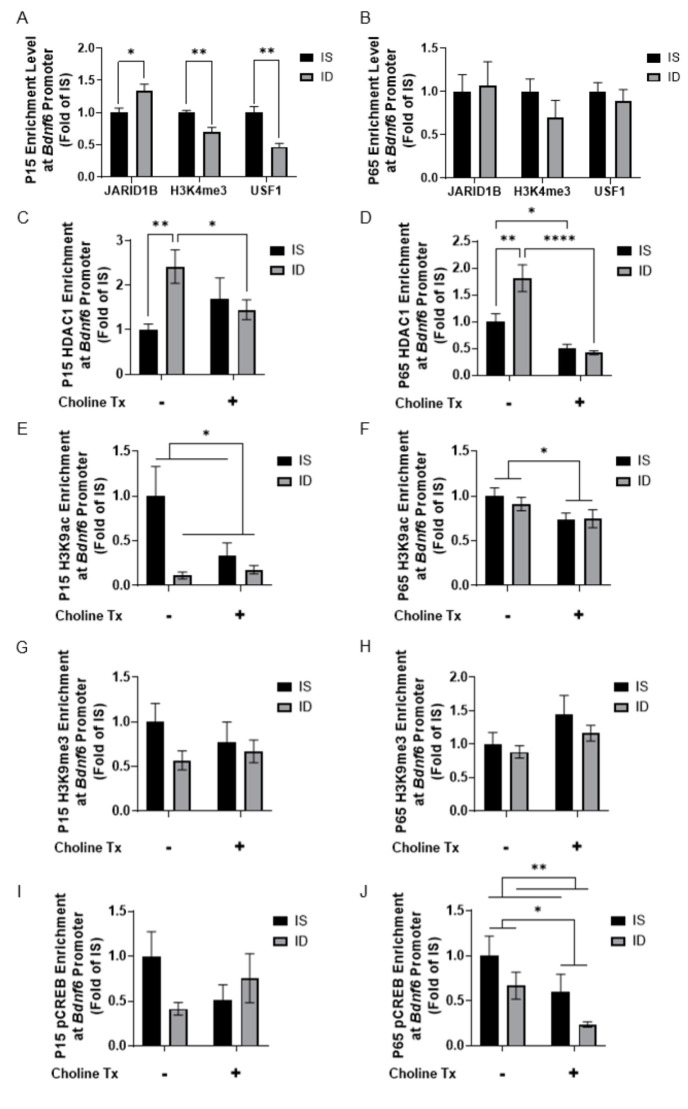
Iron deficiency and choline supplementation induce epigenetic changes at the hippocampal *Bdnf6* promoter. (**A**,**B**) Enrichment levels of JARID1B, H3K4me3 and USF1 at the *Bdnf6* promoter in P15 (**A**) and P65 (**B**) IS and ID groups. (**C**,**D**) HDAC1 enrichment at the *Bdnf6* promoter in the 4 groups at P15 (**C**) and P65 (**D**). (**E**,**F**) H3K9ac enrichment at the *Bdnf6* promoter in the 4 groups at P15 (**E**) and P65 (**F**). (**G**,**H**) H3K9me3 enrichment at the *Bdnf6* promoter in the 4 groups at P15 (**G**) and P65 (**H**). (**I**,**J**) pCREB enrichment at the *Bdnf6* promoter in the 4 groups at P15 (**I**) and P65 (**J**). All data were assessed by qChIP and normalized to the age-matched IS group. Tx: Treatment (Supplementation). Values are mean ± SEM; n = 5–6/group; * *p* < 0.05, ** *p* < 0.01, **** *p* < 0.0001.

**Table 1 nutrients-13-04527-t001:** Two-way ANOVA results. Significant *p*-values are in bold text.

2-Way ANOVA *p*-Value Table							
Figure	Interaction	Main Effect of Iron Status	Main Effect of Choline Supplementation	Simple Effect for Iron in Non-Choline-Supplemented Group	Simple Effect for Iron in Choline-Supplemented Group	Simple Effect for Choline Supplementation in Iron-Deficient Group	Simple Effect for Choline Supplementation in Iron-Sufficient Group
2A	0.1106	0.6871	0.9212	-	-	-	-
2B	**0.0341**	-	-	**0.0003**	0.2456	0.7278	**0.0096**
2C	**0.0313**	-	-	**0.0029**	0.9365	**0.0051**	0.9858
3A	0.5159	0.9654	0.1919	-	-	-	-
3B	**0.0012**	-	-	**0.0194**	**0.0086**	**0.0363**	**0.0051**
3C	0.9973	0.0978	0.1032	-	-	-	-
3D	0.1635	0.8901	0.5326	-	-	-	-
3E	**0.0331**	-	-	**0.0209**	0.4725	0.5818	**0.0116**
3F	0.6643	0.7063	**0.0014**	-	-	-	-
3G	0.4894	0.1690	0.5148	-	-	-	-
3H	0.1633	0.1536	0.9047	-	-	-	-
3I	0.5103	0.3182	**0.0079**	-	-	-	-
4A	**0.0491**	-	-	**0.0361**	0.4602	0.5953	**0.0274**
4B	0.2016	**0.0002**	0.0577	-	-	-	-
4C	0.1726	0.2513	**<0.0001**	-	-	-	-
4D	**0.0165**	-	-	**0.0059**	0.5426	**0.0112**	0.3749
4E	0.3664	**0.0025**	**<0.0001**	-	-	-	-
4F	**0.0152**	-	-	0.2469	**0.0191**	**0.0012**	0.8812
4G	0.6154	0.2673	0.3485	-	-	-	-
4H	0.1966	0.0562	0.5823	-	-	-	-
4I	**0.0081**	-	-	**0.0004**	0.9257	**0.0003**	0.7840
5C	0.1556	**0.0140**	0.1612	-	-	-	-
5D	0.5868	0.6225	**0.0190**	-	-	-	-
5E	**0.0417**	-	-	**0.0020**	0.4898	0.6952	**0.0037**
5F	0.9667	0.0557	**0.0142**	-	-	-	-
6C	**0.0199**	-	-	**0.0073**	0.6004	**0.0441**	0.1584
6D	**0.0063**	-	-	**0.0012**	0.6758	**<0.0001**	**0.0293**
6E	0.0638	**0.0114**	0.1402	-	-	-	-
6F	0.5868	0.6225	**0.0190**	-	-	-	-
6G	0.3614	0.1400	0.7255	-	-	-	-
6H	0.6555	0.2684	0.0563	-	-	-	-
6I	0.0793	0.3273	0.8499	-	-	-	-
6J	0.9143	**0.0499**	**0.0231**	-	-	-	-

## Data Availability

Not applicable.

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
