# Peer review of "Prenatal Iron Deficiency and Choline Supplementation Interact to Epigenetically Regulate Jarid1b and Bdnf in the Rat Hippocampus into Adulthood"

_nutrients, 2021, doi:10.3390/nu13124527_

Round 1

Reviewer 1 Report

In the manuscript Prenatal iron deficiency and choline supplementation interact to epigenetically regulate Jarid1b and Bdnf in the rat hippocampus into adulthood, the authors utilize a mixture of iron deficient, iron sufficient, and choline supplemented rats to study changes in the epigenetic landscape of the brain of developing rats, in order to investigate the cause of long-term cognitive impairments caused by early-life iron deficiencies.  The authors demonstrate a number of significant changes in protein expression and epigenetics markers in response to these various conditions.  Generally speaking, the manuscript is well written and the experiments seem to be well designed.  The results are explained in detail and analyzed appropriately, without overstating any potential findings.  My only real issue with the manuscript is more of a question I would like answered, and that is: in Figure 2E, why weren’t the choline supplemented groups assessed for JARID1 enzymatic activity?

Reviewer 2 Report

This is a clearly written manuscript describing complex and detailed but logically sequenced studies of a potential mechanism by which prenatal iron deficiency may exert its adverse effects on neurodevelopment. The authors have identified an additional area for investigation, in that the potential beneficial effects of calling appear to change at different stages of development and iron status.

My only suggested modification is a very minor one. In the Discussion at the start of each section, the authors have inserted a brief sentence stating the point that will be demonstrated in the section that follows. It would be helpful to the reader if that statement was either underlined, in italics, or in bold print

Author Response

Please see attached response
